# Cellular repressor of E1A-stimulated genes 1 enhances skeletal muscle performance through the stimulation of muscle differentiation and Akt-mTOR signaling pathway activation

Ayumi Goto[1,2]*, Michihiro Hashimoto[3], Sho Yokogawa[2], Yuzu Naruse[2], Hitoshi Yamashita[2]

**1** Department of Physiological therapy, School of Health Science, Toyohashi SOZO University, Toyohashi, Japan, **2** Department of Biomedical Sciences, College of Life and Health Sciences, Chubu University, Kasugai, Japan, **3** Division of Advanced Medical Science, Asahikawa Medical University, Asahikawa, Japan

\* a-goto@sozo.ac.jp

## Abstract

Cellular repressor of E1A-stimulated genes 1 (CREG1), a glycoprotein secreted by various cell types, plays a crucial role in cellular differentiation and energy metabolism. While previous research has linked CREG1 deficiency in skeletal muscles to impaired exercise capacity and altered muscle fiber-type composition, its specific role in skeletal muscle function and differentiation remains unclear. In this study, we investigated the impact of CREG1 on muscle performance and fiber-type composition in adipocyte P2-CREG1-transgenic (Tg) mice and explored muscle differentiation in C2C12 myotubes. Tg mice exhibited significantly improved muscle performance compared to wild-type mice, as indicated by enhanced grip strength. Additionally, the proportion of type IIx fiber in the soleus muscle was significantly increased in Tg mice, along with a tendency towards elevated *Myh1* mRNA expression. Enhanced CREG1 expression and activation of the Akt-mTOR signaling pathway, which is involved in muscle protein synthesis, were observed in the skeletal muscles of Tg mice. In C2C12 myotubes, *Creg1* knockdown appears to decrease myoblast determination protein 1 (*Myod1*) expression, while recombinant CREG1 treatment restored *Myod1* expression and promoted Akt-mTOR phosphorylation. These findings suggest that CREG1 stimulates muscle differentiation by enhancing protein synthesis, thereby influencing skeletal muscle function.

## Introduction

In humans, skeletal muscle constitutes 40–50% of total body mass and plays a crucial role in various physiological processes, including movement, posture, and metabolism [1,2]. Sarcopenia refers to the age-related decline in muscle mass, strength,

**Funding:** Grant-in-Aid for Scientific Research (Kakenhi) from the Japan Society for the

Promotion of Science (20K06450 and 23K10881 to Hitoshi Yamashita, 21K17668 to Ayumi Goto, 18K11034 to Michihiro Hashimoto), YOKOYAMA Foundation for Clinical Pharmacology (YRY-2020 to Ayumi Goto), Naito Research Grant (Ayumi Goto), The Hibi Science Foundation (Ayumi Goto), Chubu University Grant R (Ayumi Goto), and Toyohashi Sozo University research grant (Ayumi Goto). The funders had no role in study design, data collection and analysis, decision to publish, or preparation of the manuscript.

**Competing interests:** There are no competing interests to declare.

and function. Sarcopenia is a key contributor to the development of locomotive syndrome, which is associated with physical disability and poor quality of life [3,4]. Furthermore, sarcopenia is associated with an elevated risk of cardiovascular disease and type 2 diabetes mellitus [5]. Therefore, maintaining skeletal muscle mass and strength is essential for preventing various diseases and maintaining overall health.

Cellular repressor of E1A-stimulated genes 1 (CREG1) is a secreted glycoprotein consisting of 220 amino acids (aa), including a 31-aa signal peptide in both humans and mice. Initially identified as a transcriptional regulator, CREG1 binds to the retinoblastoma tumor suppressor protein *in vitro* and represses the E1A-mediated activation of the adenovirus E2 promoter. In addition to its role in transcriptional regulation, CREG1 inhibits cell growth and promotes the differentiation of several cell types, including human embryonic teratocarcinoma NTERA-2 [6], smooth muscle [7], and cardiomyogenic [8] cells. Our previous study demonstrated that CREG1 upregulates uncoupling protein 1 and promotes brown adipogenesis in the murine mesenchymal stem cell line C3H10T1/2 [9]. An examination of the effects of CREG1 on energy metabolism in adipocyte P2 (aP2)-CREG1-transgenic (Tg) mice revealed that CREG1 overexpression stimulates brown fat thermogenesis and prevents diet-induced obesity [10]. Furthermore, administration of the recombinant form of CREG1 stimulates the differentiation of brown adipocytes and ameliorates diet-induced obesity in mice [11]. Recent findings from our group further demonstrated an increase in CREG1 expression, with the concurrent induction of myoblast determination protein 1 (MyoD), a key marker of skeletal muscle differentiation, during muscle regeneration [12]. However, the role of CREG1 in skeletal muscle differentiation remains incompletely understood.

Consequently, determining the role of CREG1 in skeletal muscle has gained considerable attention. In a study involving muscle-specific CREG1-knockout (KO) mice, Song *et al.* (2021) reported that the absence of CREG1 in skeletal muscles reduced the anti-fatigue capacity during endurance exercise in 9-month-old mice [13]. Muscle performance is closely linked to muscle fiber-type composition, which is broadly categorized into two types: (1) fast-twitch fibers, which are optimized for explosive power and force generation, and (2) slow-twitch fibers, which are adapted for endurance in sustained low-intensity exercise [1]. Notably, CREG1 deficiency resulted in a reduced proportion of type I fibers and an increased proportion of type II fibers in the skeletal muscles of 9-month-old muscle-specific CREG1-KO mice [13]. This finding suggests that CREG1 may influence skeletal muscle performance and muscle fiber-type composition. However, the effects of CREG1 on skeletal muscle strength and fiber-type composition remain incompletely understood.

Akt (protein kinase B) is a serine/threonine-specific protein kinase that plays a key role in various cellular processes, including protein synthesis, glucose metabolism, apoptosis, and cell proliferation [14]. The Akt/ mechanistic target of rapamycin (mTOR) pathway is essential for integrating intracellular signaling related to protein synthesis in skeletal muscle cells [15]. In addition to regulating muscle protein synthesis, the Akt/mTOR pathway also plays a critical role in muscle differentiation and regeneration [16–18]. In our previous study, we observed a significant increase in

CREG1 expression during muscle regeneration. However, there is currently no evidence of the impact of CREG1 on the Akt/mTOR signaling pathway in skeletal muscle.

Therefore, in the present study, using aP2-CREG1-Tg mice, we aimed to determine the effect of CREG1 on skeletal muscle phenotypes, including muscle strength and fiber-type composition. This approach was based on the fact that skeletal muscle and brown adipocytes share the same embryological origin [19,20]. Additionally, we evaluated the effect of CREG1 on skeletal muscle differentiation using mouse C2C12 myoblast cultures.

## Materials and methods

### Animals

The aP2-CREG1-Tg (Tg) mice were generated as previously described [10]. The aP2 protein is selectively expressed in adipocytes, as well as in macrophages [21,22]. We crossed the heterozygous transgenic mice (Tg; 14 th generation) with their wild-type (WT) littermates, and used the resulting Tg mice and their WT littermates for the experiments. Male WT and Tg mice (line 52) were fed a standard chow diet (CE-2; CLEA Japan, Inc., Shizuoka, Japan), and tap water was provided *ad libitum*. These mice were housed at approximately 23 °C under a 12-h light/12-h dark cycle. At 5 months of age, blood and tissue samples were collected from the mice under isoflurane anesthesia and stored at −80 °C. All experiments were conducted in strict accordance with the recommendations outlined in the Fundamental Guidelines for Proper Conduct of Animal Experiment and Related Activities in Academic Research Institutions under the jurisdiction of the Ministry of Education, Culture, Sports, Science, and Technology, Japan. The protocol was approved by the Institutional Animal Care and Use Committee of Chubu University (#202110004). The study was carried out in accordance with the ARRIVE guidelines.

### Grip strength test

The grip strength test was performed following our previously established protocol [23]. Grip strength was measured using a force transducer (Model-761; AIKOH ENGINEERING, Aichi, Japan). Briefly, 3-month-old mice were allowed to rest on a horizontal wire mesh with their forelimbs and then gently pulled back until their maximum muscle strength was reached. A total of five consecutive measurements were recorded, and the mean grip strength was calculated, excluding the maximum and minimum values. The data, normalized to body weight, are expressed as N/g.

### Immunohistochemistry (IHC) analyses

Serial transverse cryosections (8-μm thick) of the frozen samples were cut at −20 °C and mounted on glass slides [24]. The sections were air-dried and then blocked with 1% Roche Blocking Regent (Roche Diagnostics, Penzberg, Germany) for 1 h at 23 °C. After blocking, the samples were incubated overnight at 4 °C with primary antibodies targeting type I [1:100; BA-D5, Developmental Studies Hybridoma Bank (DSHB), Iowa City, IA, USA], type IIa (1:200; SC-71, DSHB), type IIb (1:100; BF-F3, DSHB) fibers, and laminin (1:200; L9393, Sigma-Aldrich, St. Louis, MO, USA) [25]. Staining was visualized by incubating the sections with the following secondary antibodies: goat anti-mouse IgG2b cross-adsorbed secondary antibody, Alexa Fluor™ 350 (1:500; A21140, Invitrogen, Carlsbad, CA, USA); goat anti-mouse IgG1 cross-adsorbed secondary antibody, Alexa Fluor™ 488 (1:500; A21121, Invitrogen); goat anti-mouse IgM cross-adsorbed secondary antibody, Alexa Fluor™ 555 (1:500; A21426, Invitrogen); and goat anti-rabbit IgG (H + L) cross-adsorbed secondary antibody, Alexa Fluor™ 647 (1:500; A21244, Invitrogen) for 1h at room temperature. Samples were visualized under a microscope (BZ-9000, Keyence, Osaka, Japan) and analyzed using Image J software for IHC staining. All stained fibers in the muscle cross-section were counted manually.

### C2C12 cell culture

Cell culture was performed as previously described [26,27]. Mouse myoblast C2C12 cells were cultured in 12-well plates with type I collagen-coated surfaces [AGC techno glass (IWAKI), Shizuoka, Japan]. The cells were maintained in

 

Dulbecco's modified Eagle's medium (DMEM), supplemented with 10% heat-inactivated fetal bovine serum, high glucose (4,500 mg glucose/L), 4 mM L-glutamine, 110 mg/L sodium pyruvate, and 1% Penicillin-Streptomycin Solution (168–23191, WAKO, Osaka, Japan) in a humidified atmosphere of 95% air and 5% $CO_2$ at 37 °C. When the cells reached approximately 80% confluence, the culture medium was replaced with a differentiation medium (DMEM supplemented with 2% horse serum, low glucose (1,000 mg glucose/L),4 mM L-glutamine, and 1% Penicillin-Streptomycin Solution). The differentiation medium was replaced with a fresh medium every 2 days.

## RNA interference and CREG1 treatment

Twenty-four hours after seeding or 3 days after differentiation, RNA oligos were transfected into myoblasts or myotubes using Lipofectamine RNAiMax reagent (Thermo Fisher Scientific, Waltham, MA, USA), as previously described [12]. *Creg1*-silencing experiments were performed using stealth small interfering RNA (siRNA) targeting CREG1 (MSS243209) and a scrambled stealth negative control (Invitrogen). The final siRNA concentration was 5 nM. One day after transfection, the cells were stimulated with purified C-terminal His-tag-fused CREG1 (CREG1-MH), as previously described [12]. CREG1-MH was added to the differentiation medium at a final concentration of 1 μg/mL.

## Quantitative reverse transcription-polymerase chain reaction

Total RNA was extracted using TRIzol reagent (Thermo Fisher Scientific) and reverse-transcribed using a High-Capacity cDNA Reverse Transcription Kit (Thermo Fisher Scientific), according to the manufacturer's instructions [28,29]. Quantitative reverse transcription-polymerase chain reaction (qRT-PCR) was performed using a Light-Cycler® and THUNDER-BIRDTM SYBR® qPCR Mix (TOYOBO, Osaka, Japan) to quantify mRNA levels. Gene expression was quantified using the relative standard curve method, with *36b4* serving as an internal standard for the normalization of the total RNA amount in each reaction. The following primers were used: *Creg1*, 5′-GACCTGCAGGAAAATCCAGA-3′ (forward) and 5′-AACAAA CAGCGAATCCCTTG-3′ (reverse); *Myh1*, 5′-AGTCCCAGGTCAACAAGCTG-3′ (forward) and 5′-CACATTTTGCT CATCTCTCTTTG-3′ (reverse); *Myh2*, 5′-AGTCCCAGGTCAACAAGCTG-3′ (forward) and 5′-GCAT GACCAAAGGTTTCACA-3′ (reverse); *Myh4*, 5′-AGTCCCAGGTCAACAAGCTG-3′ (forward) and 5′-TTTCTCCTGT CACCTCTCAACA-3′ (reverse); *Myh7*, 5′-AGTCCCAGGTCAACAAGCTG-3′ (forward) and 5′-TTCCACCTAAAGGGCT GTTG-3′ (reverse); *Myod1*, 5′-AAGACGACTCTCACGGCTTG-3′ (forward) and 5′-GCAGGTCTGGTGAGTCGAAA-3′ (reverse); and *36b4*, 5′-TCATCCAGCAGGTGTTTGACA-3′ (forward) and 5′- CCCATTGATGATGGAGTGTGG-3′ (reverse).

## Western blot analysis

Western blot analysis was performed as previously described [30,31]. Protein samples (10 μg) were separated using sodium dodecyl sulfate-polyacrylamide gel electrophoresis (SDS-PAGE) on a 12% polyacrylamide gel and transferred onto polyvinylidene difluoride membranes (Millipore, Burlington, MA, USA). The membranes were blocked for 1 h at room temperature in nonfat dry milk and then incubated overnight at 4 °C with the following antibodies: anti-CREG1 (ab233282; Abcam, Cambridge, UK), anti-Akt Ser[473] [9271; Cell Signaling Technology (CST), Danvers, MA, USA], anti-Akt (9272; CST), anti-glyceraldehyde-3-phosphate dehydrogenase (GAPDH) (2118; CST), anti- insulin-like growth factor 2 receptor (IGF2R) (14364; CST), anti- mTOR Ser[2448] (2971; CST), anti- mTOR (2972; CST), and anti-α/β-tubulin (2148; CST). After incubation with secondary antibodies for 1 h at room temperature, protein bands were visualized using ImmunoStar LD (WAKO). Protein signals were detected using the LAS4000 system (FUJIFILM, Tokyo, Japan).

## Statistical analyses

Data are expressed as mean ± standard error of the mean (SEM). Multiple comparisons were made using a one-way ANOVA, followed by post-hoc analysis with the Tukey–Kramer test or Student's *t*-test. Statistical significance was set at $P < 0.05$.

## Results

### aP2-CREG1-Tg mice exhibited elevated muscle performance

In a previous study, we demonstrated that CREG1 plays an important role in muscle regeneration [12]. In the present study, to evaluate the effect of CREG1 on muscle performance, we first assessed muscle strength in Tg mice using the grip strength test. There were no significant differences in body weight or muscle mass between Tg and littermate WT mice (Table 1). However, the grip strength of the forelimbs in Tg mice was significantly higher (13%) than that in WT mice (Fig 1). This finding suggests that Tg mice exhibit superior muscle function relative to WT mice.

### CREG1 upregulation in skeletal muscle alters the composition of muscle fiber types

Muscle strength is closely related to fiber-type composition [32,33]. Skeletal muscle fibers are broadly classified into "slow-twitch" (type I) and "fast-twitch" (type II) fibers, with type II muscle fibers designed for short, powerful bursts of energy. To investigate potential changes in muscle-fiber phenotypes in Tg skeletal muscles, we assessed fiber-type composition using immunohistochemistry staining of the soleus and plantaris muscles in WT and Tg mice. In the soleus muscle, the proportion of type IIx muscle fibers was higher in Tg mice than in WT mice (Figs 2A and 2C). Consistently, the number of type IIx fibers in the soleus muscle of Tg mice was significantly higher than that in WT mice (Fig 2E). In contrast, no significant differences in fiber-type distribution were observed in the plantaris muscle between WT and Tg mice (Figs 2B and 2D). Similarly, there was no significant difference in the number of fiber types in the plantaris muscle between WT and Tg mice (Fig 2F). Skeletal muscle contains four myosin heavy-chain isoforms: Myh7 (Type I), Myh2 (Type IIa), Myh1 (Type IIx) and Myh4 (Type IIb). The *Myh1* and *Myh7* mRNA levels tended to be higher in the soleus muscles of Tg mice compared to WT mice, although the differences were not statistically significant (Fig 2G). In contrast, the levels of myosin heavy-chain isoforms in the plantaris muscle did not differ between the groups (Fig 2H).

### CREG1 upregulation in skeletal muscle tissue enhances Akt-mTOR signaling

We examined the mRNA and protein expression of CREG1 in the soleus muscles of mice, as CREG1 expression is significantly induced during muscle regeneration [12]. Although the serum CREG1 level was higher in Tg mice than in WT mice (Fig 3A), the difference was not statistically significant. However, the *Creg1* mRNA level in the soleus muscle was significantly higher in Tg mice than in WT mice (Fig 3B). Similarly, the CREG1 protein level in the soleus muscle was significantly higher in Tg mice than in WT mice (Fig 3C). A previous study demonstrated that CREG1 binds to IGF2R [34], and in the soleus muscle, IGF2R expression tended to be higher in Tg mice than in WT mice (Fig 3D and 3E). Skeletal muscle development is regulated at the translational level through the stimulation of protein synthesis [35]. Therefore, we investigated the effect of CREG1 on Akt-mTOR signaling [36]. Notably, phosphorylation of Akt Ser$^{473}$ (Fig 3D and 3E) and mTOR Ser$^{2442}$ (Fig 3D and 3E) in the soleus muscles was significantly higher in Tg mice than in WT mice, suggesting an

**Table 1. Body weight and muscle weight in wild-type (WT) and adipocyte P2-CREG1-transgenic (Tg) mice.**

|  | WT | Tg | *p*-value |
|---|---|---|---|
| **Body Weight (g)** | 31.0±0.63 | 29.6±0.67 | 0.197 |
| **Absolute muscle wet weight** |  |  |  |
| **Soleus (mg)** | 11.0±0.55 | 9.2±0.30 | 0.065 |
| **Plantaris (mg)** | 21.0±0.94 | 19.4±0.66 | 0.209 |
| **Relative muscle wet weight** |  |  |  |
| **Soleus (mg/g)** | 0.35±0.01 | 0.31±0.01 | 0.057 |
| **Plantaris (mg/g)** | 0.68±0.03 | 0.65±0.009 | 0.441 |

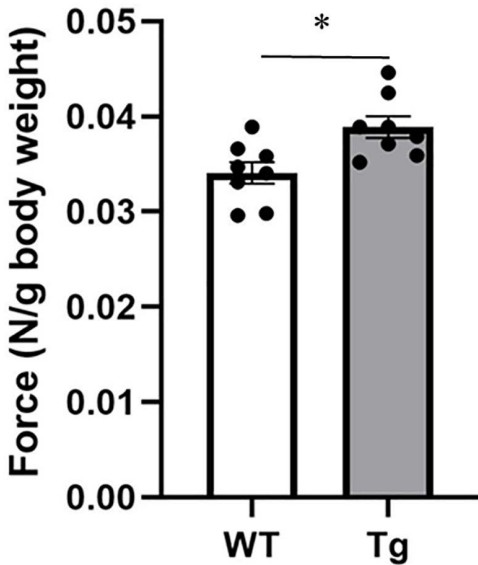

**Fig 1. Muscle performance in wild-type (WT) and adipocyte P2-CREG1-transgenic (Tg) mice.** Grip strength was performed on 3-month-old WT and Tg mice. Forelimb grip strength. Data are presented as mean ± SEM; n = 8 per group. A student's *t*-test was performed; *$P < 0.05$, versus WT.

increase in muscle protein synthesis in Tg mice. There was no significant difference in total Akt and mTOR levels between WT and Tg mice (Fig 3D and 3E).

### Effects of CREG1 on muscle differentiation and Akt-mTOR signaling *in vitro*

To further investigate the role of CREG1 in muscle development, we examined its impact on the differentiation of mouse C2C12 myoblasts into myotubes *in vitro*. The *Creg1* mRNA level increased significantly over the course of 1–3 days of differentiation stimulation in C2C12 cells (Fig 4A). MyoD, a key transcriptional regulator of muscle differentiation in C2C12 cells [37], exhibited a delayed upregulation of *Myod1* mRNA expression (Fig 4B) compared to the early increase in *Creg1* mRNA expression during differentiation. Next, we determined whether CREG1 influences *Myod1* expression during C2C12 cell differentiation. One day after *Creg1* knockdown in myoblasts, the cells were treated with C-terminal His-tag-fused CREG1 (CREG1-MH) for 4 days in a differentiation medium. The expression of *Creg1* mRNA was significantly suppressed in myotubes transfected with *Creg1* siRNA compared to those transfected with scrambled non-targeting control siRNA (Con siRNA) (Fig 4C). As shown in Fig 4D, the *Myod1* mRNA level shows a tendency to decrease in myotubes transfected with *Creg1* siRNA compared to those transfected with Con siRNA; however, no significant difference was observed. Notably, this decrease in *Myod1* mRNA was reversed upon treatment with recombinant CREG1-MH.

Finally, we evaluated the effects of CREG1 on the protein synthesis pathway in C2C12 cells. Myotubes treated with *Creg*1 siRNA and CREG1-MH showed a time-dependent increase in CREG1 protein levels (Fig 5A), consistent with findings from our previous study in C2C12 myotubes [12]. As shown in Figs 5B and 5C, phosphorylation of Akt Ser$^{473}$ and mTOR Ser$^{2448}$ significantly increased upon CREG1-MH treatment in *Creg1* siRNA-transfected myotubes in a time-dependent manner. At 24 h, phosphorylation of Akt and mTOR was 40.6- and 7.0-fold higher than that at 0 min, respectively (Figs 5B and Fig 5C).

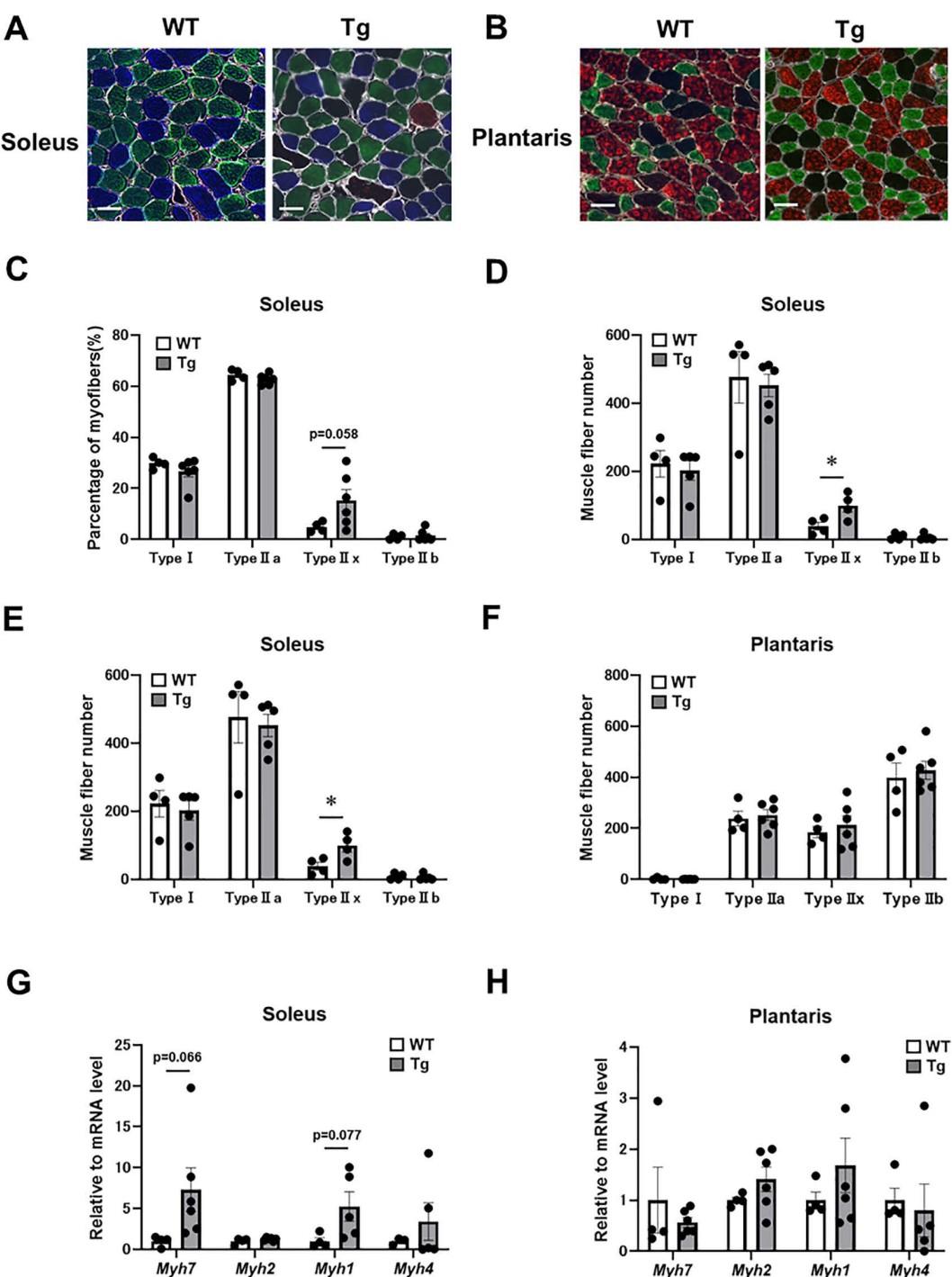

**Fig 2. Muscle fiber-type composition and myosin heavy chain isoforms in wild-type (WT) and adipocyte P2-CREG1-transgenic (Tg) mice.**
Muscle fiber-type composition was analyzed in the soleus and plantaris muscles of 5-month-old WT and Tg mice. (A-B) Immunostaining of myofibers in the soleus (A) and plantaris (B) muscles of WT and Tg mice. Type I (blue), type IIa (green), type IIx (black), type IIb (red), and lamin (white). Scale bar, 50 μm. (C-D) Distribution of muscle fiber types in the soleus (C) and plantaris (D) muscles. (E-F) Number of myofibers in the soleus (E) and plantaris (F) muscles of WT and Tg mice. (G-H) Relative levels of *Myh7*, *Myh2*, *Myh1* and *Myh4* mRNA in the soleus (G) and plantaris (H) muscles. Data were normalized to the *36b4* mRNA level. Data are presented as mean ± SEM; n = 4–6 per group. A student's *t*-test was performed; *$P < 0.05$, versus WT.

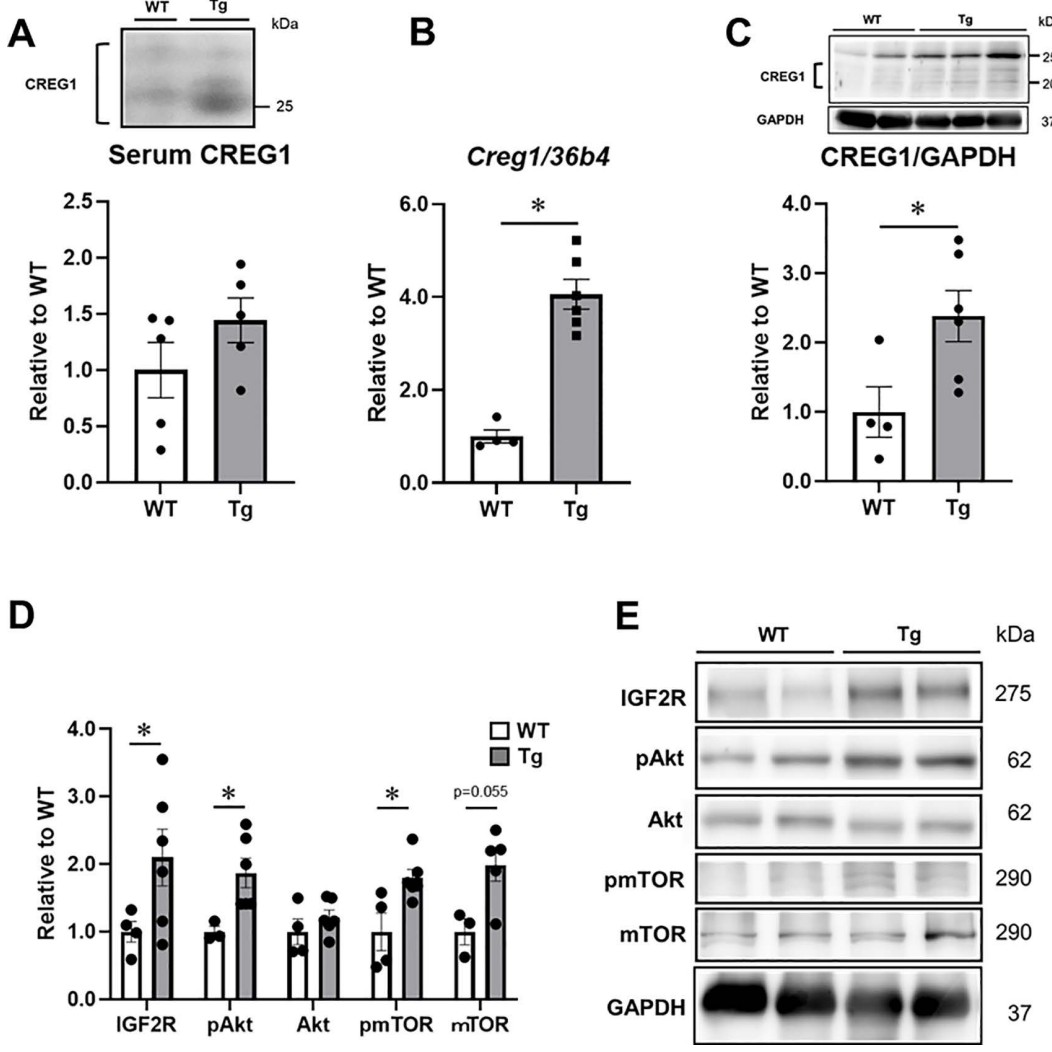

**Fig 3. Effect of CREG1 on protein expression in the soleus muscles of adipocyte P2-CREG1-transgenic (Tg) mice.** (A) Western blotting for CREG1 was performed using the serum (1 μL) of wild type (WT) and Tg mice. (B-C) The expression of CREG1 in the soleus muscles of WT and Tg mice was examined using RT-PCR and western blot analysis. Relative levels of *Creg1* mRNA (B) and CREG1 protein (C) in the soleus muscles. (D) Relative levels of insulin-like growth factor 2 receptor (IGF2R), phosphorylated Akt Ser$^{473}$ (pAkt), total Akt (Akt), phosphorylated mechanistic target of rapamycin (pmTOR) Ser$^{2448}$ and total mTOR (mTOR) in the soleus muscles. Data were normalized to those of glyceraldehyde-3-phosphate dehydrogenase (GAPDH). (E) Western blotting for IGF2R, pAkt, Akt, pmTOR, mTOR and GAPDH in the soleus muscles. Representative images are shown. Data are presented as mean ± SEM; n = 4–6 per group. A student's *t*-test was performed; *$P < 0.05$, versus WT.

## Discussion

This study revealed significant enhancement of muscle performance in aP2-CREG1-Tg mice, as demonstrated by increased grip strength. These results suggest that CREG1 may contribute to the observed improvements in muscle strength in the Tg mice. Consistent with our findings, Song *et al.* (2021) recently reported that skeletal muscle-specific CREG1-KO mice exhibited a significant reduction in exercise duration to exhaustion and that recombinant CREG1 protein administration improved muscle motor function in CREG1-KO mice [13]. Furthermore, our results showed a significant increase in CREG1 expression in the soleus muscle of aP2-CREG1-Tg mice compared to WT mice (Fig 3C). These

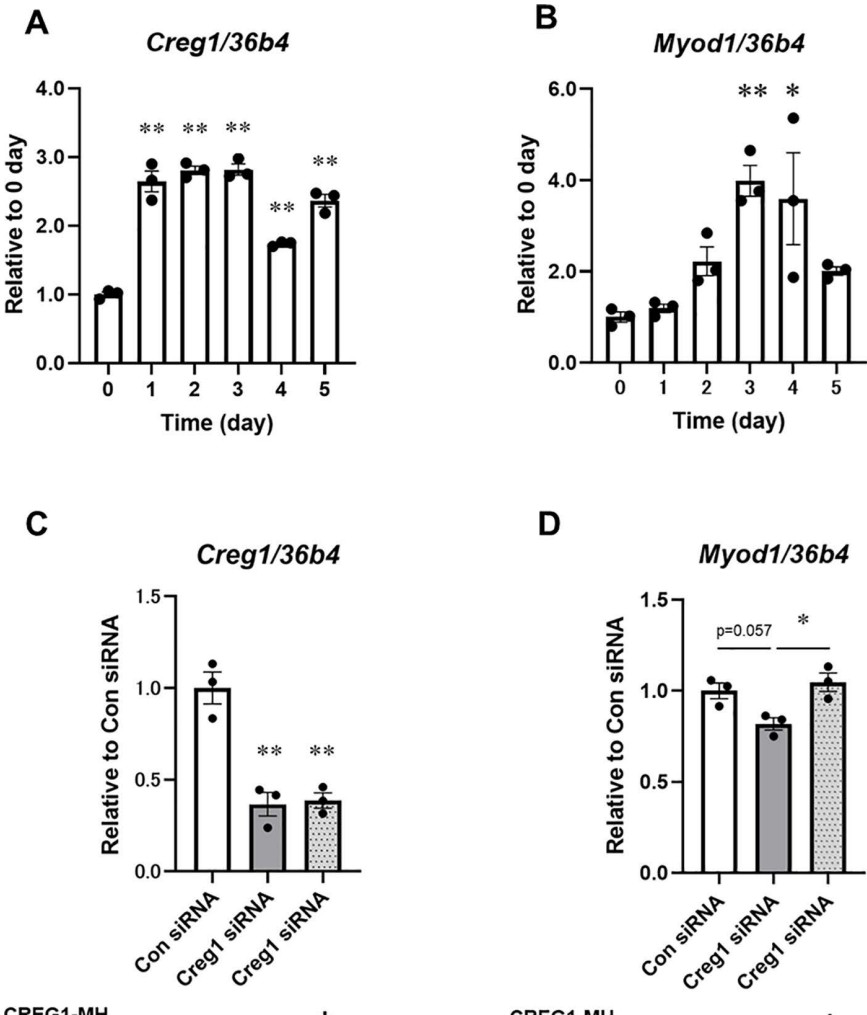

**Fig 4. Effect of CREG1 treatment on gene expression during muscle differentiation.** (A-B) Time course of *Creg1* (A)and *Myod1* (B) mRNA expression in C2C12 myotubes. Mouse C2C12 cells were differentiated into myotubes for 5 days. (C-D) The mRNA level of *Creg1* (C) and *Myod1* (D) in siRNA-transfected C2C12 cells with or without CREG1. C2C12 cells were transfected with scrambled non-targeting control siRNA (Con) or *Creg1*-specific siRNA and then treated with recombinant CREG1 (1 µg/mL) for 4 days. The data were normalized to *36b4* level. The data are presented as mean ± SEM; n = 3 per group. One-way ANOVA with Tukey–Kramer multiple comparisons post hoc test was performed; *$P < 0.05$, **$P < 0.01$, versus 0 day or Con siRNA.

findings suggest that the increase in muscle strength in aP2-CREG1-Tg mice may be associated with elevated expression of CREG1 in skeletal muscle.

Fast-twitch fibers utilize anaerobic metabolism to generate fuel, making them more adept at producing short bursts of strength or speed than slow-twitch fibers [1]. Consequently, athletes, particularly sprinters, tend to have a higher proportion of fast-twitch fibers than non-athletes [38]. Fast-twitch fibers are glycolytic, using anaerobic glycogen breakdown for energy production required for high-intensity, short-duration activities [1,39]. Accordingly, this is reflected in the significantly higher anaerobic capacity observed in athletes, as demonstrated by the Wingate test [40]. In muscle-specific CREG1-KO mice, *Myh1* mRNA expression was significantly reduced in the gastrocnemius muscle, which contains both slow- and fast-twitch fibers [13]. This finding aligns with our results and suggests that CREG1 plays a role in promoting

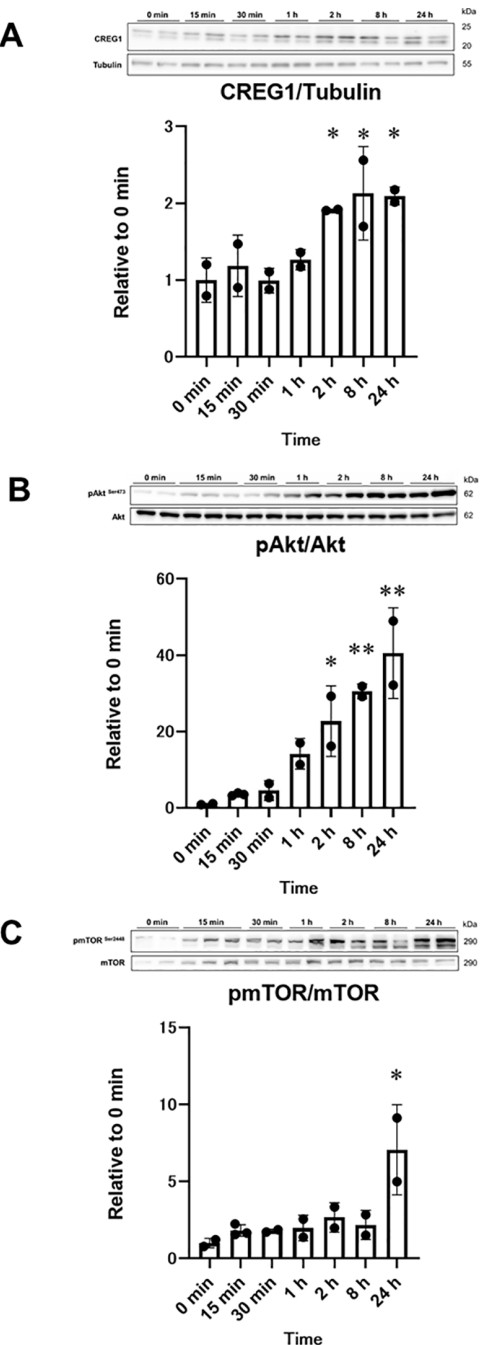

**Fig 5. Effect of CREG1 treatment on protein expression in myotubes.** Mouse C2C12 cells transfected with *Creg1*-specific siRNA were differentiated into myotubes and then treated with recombinant CREG1 (1 μg/mL) for the indicated time. (A) Relative levels of CREG1 in C2C12 myotubes (n = 2–3). The data were normalized to Tubulin level. (B) Relative levels of phospho-Akt Ser473 in C2C12 myotubes (n = 2–3). The data were normalized to the total Akt level. (C) Relative levels of phospho-mechanistic target of rapamycin (mTOR) Ser2448 in C2C12 myotubes (n = 2–3). The data were normalized to the total mTOR level. The data are presented as mean ± SEM. One-way ANOVA with Tukey–Kramer multiple comparisons post hoc test was performed; *$P$ < 0.05, **$P$ < 0.01, versus 0 min.

type IIx fibers. However, another study using CREG1-KO mice reported a decrease in the proportion of type I fibers and an increase in type II fibers in the soleus muscle, although they did not assess *Myh* gene expression [13]. Notably, while KO mice showed an increase in type II fibers, our study found an increase in the proportion of type IIx fibers in the Tg-soleus muscle, which may explain the discrepancy. The study by Song *et al.* (2021) did not categorize the type II fibers, which could account for the differences in our results. However, the reasons for the differences between our study findings and those reported by Song *et al.* (2021) remain incompletely understood. One possible explanation could be due to the source of CREG1. Because CREG1 is thought to be mainly supplied by the liver to other organs, including skeletal muscles. Therefore, exogenous CREG1 produced by non-muscle tissues may still influence skeletal muscle function in the absence of endogenous CREG1 in skeletal muscle. Indeed, exogenous CREG1 is incorporated into myotubes via IGF2R and stimulates cellular glucose uptake [12]. Therefore, it is possible that exogenous CREG1 affects muscle biology through IGF2R even in the absence of endogenous CREG1. Further studies are needed to fully understand the changes in *Myh* expression patterns and classification of type II fibers in the soleus muscle of muscle-specific CREG1-KO mice.

In the present study, we observed a significant increase in *Creg1* mRNA levels in the skeletal muscles of aP2-CREG1-Tg mice compared to WT mice (Fig 3B). This observation was unexpected because the aP2-CREG1-Tg mice were specifically designed to overexpress CREG1 in adipose tissue. Indeed, *Creg1* mRNA expression in the adipose tissue of Tg mice was approximately 20-fold higher than in the adipose tissue of WT mice, while no induction was detected in the liver, heart, or kidney [10]. In contrast, the increase in *Creg1* mRNA levels in the soleus muscle was considerably lower (4-fold) than in the adipose tissues of Tg mice. The exact reason for the stimulation of *Creg1* expression in skeletal muscles remains unclear. However, a possible explanation is that CREG1 secreted from adipose tissue and the liver may circulate through the bloodstream and influence skeletal muscle. Although serum CREG1 levels were higher in Tg mice than in WT mice, no significant differences were observed (Fig 3A). This observation suggests that the effects of CREG1 on skeletal muscle could be mediated not only by exogenous CREG1 but also by endogenous CREG1 in an autocrine and/or paracrine manner. In support of this hypothesis, CREG1 administration stimulates the differentiation of brown adipocytes [11], which share the same embryological origin as skeletal muscles [19,20]. Moreover, *Creg1* expression is significantly induced during muscle regeneration in mice, which correlates with an increase in *Myod1* expression [12]. We previously demonstrated that CREG1 stimulates glucose uptake through AMPK activation in skeletal muscle cells [12]. More recently, Song *et al.* (2021) reported similar findings using a conditional mouse model, showing that CREG1 influences muscle regeneration through AMPK signaling; however, the connection between CREG1 and Akt/mTOR signaling, as well as protein synthesis, was previously unknown.

In the present study, we found that the mRNA levels of *Creg1* and *Myod1* increased concomitantly during C2C12 myoblast differentiation (Figs 4A and 4B). Moreover, our results showed that *Creg1* knockdown decreased *Myod1* expression, while recombinant CREG1 treatment restored its expression (Fig 4D), suggesting that CREG1 may regulate *Myod1* expression. Therefore, exogenous CREG1 may have stimulated its own transcription in the skeletal muscles of Tg mice, contributing to the rearrangement of muscle fiber composition. Future studies are needed to clarify the molecular mechanisms by which CREG1 stimulates fiber-type transitions in skeletal muscles.

Muscle fiber-type transition is a complex process involving protein synthesis and degradation. Autophagy, a catabolic process that degrades organelles and cytoplasmic constituents in the lysosome, has been implicated in this process [41]. Song *et al.* (2021) reported that CREG1 protects against myocardial ischemia/reperfusion injury by regulating myocardial autophagy [42], and CREG1 also plays a role in mitophagy during growth or disease development [13]. In the present study, we observed a trend toward lower soleus muscle weight in Tg mice than in WT mice, suggesting that CREG1 may stimulate autophagy for muscle rearrangement. In contrast, the Akt-mTOR-p70S6K pathway is widely recognized as a key regulator of protein synthesis [36]. Wilson *et al.* (2007) demonstrated that Akt1 activation positively regulates MyoD expression and myogenic differentiation [43]. Additionally, studies using rapamycin, an mTOR inhibitor, have shown that inhibition of mTOR negatively affects C2C12 myogenesis, indicating the importance of the Akt-mTOR pathway in muscle

differentiation [44]. In the present study, we observed significant activation of the Akt-mTOR pathway in the skeletal muscles of Tg mice, as well as in C2C12 myotubes treated with CREG1-MH. These findings suggest that CREG1 may regulate both protein synthesis and myogenic gene expression through Akt-mTOR pathway activation.

In conclusion, this study suggests that CREG1 plays an important role in skeletal muscle differentiation and function. Increased CREG1 expression may contribute to the enhancement of muscle strength. In addition, CREG1 can stimulate *Myod1* expression by activating the Akt/mTOR signaling pathway, supporting muscle differentiation.

## Limitations

In this study, we revealed that CREG1 enhances Akt-mTOR signaling in skeletal muscles; however, the underlying mechanism remains unclear. Another limitation of this study is the absence of data on muscle performance and phenotype of skeletal muscle-specific CREG1-Tg or KO models. Future studies using transgenic animals should address these gaps.

## Supporting information

**S1 Raw images. Raw data images of the original immunohistochemistry and western blot images.** (PDF)

**File S1. Supporting data: Table S1. Data for Figure 1.** Table S2. Data analysis results for Figure 1 using Student's t-test. Table S3. Data for Figure 2C. Table S4. Data analysis results for Figure 2C using Student's t-test. Table S5. Data for Figure 2D. Table S6. Data analysis results for Figure 2D using Student's t-test. Table S7. Data for Figure 2E. Table S8. Data analysis results for Figure 2E using Student's t-test. Table S9. Data for Figure 2F. Table S10. Data analysis results for Figure 2F using Student's t-test. Table S11. Data for Figure 2G. Table S12. Data analysis results for Figure 2G using Student's t-test. Table S13. Data for Figure 2H. Table S14. Data analysis results for Figure 2H using Student's t-test. S15. Data for Figure 3A-C. Table S16. Data analysis results for Figure 3A-C using Student's t-test. Table S17. Data for Figure 3D. Table S18. Data analysis results for Figure 3D using Student's t-test. Table S19. Data for Figure 4A-B. Table S20. Data analysis results for Figure 4A-B using one-way ANOVA with post-hoc Tukey–Kramer test. Table S21. Data for Figure 4C-D. Table S22. Data analysis results for Figure 4C-D using one-way ANOVA with post-hoc Tukey–Kramer test. Table S23. Data for Figure 5A-C. Table S24. Data analysis results for Figure 5A-C using one-way ANOVA with post-hoc Tukey–Kramer test.
(PDF)

## Acknowledgments

The authors would like to thank M. Matsui, Y. Yamashita, Y. Nishimoto, Y. Endo, I. Matsuda, and K. Miyake (Chubu University) for their technical support.

## Author contributions

**Conceptualization:** Ayumi Goto, Hitoshi Yamashita.

**Data curation:** Ayumi Goto.

**Formal analysis:** Ayumi Goto.

**Funding acquisition:** Ayumi Goto, Hitoshi Yamashita.

**Investigation:** Ayumi Goto, Sho Yokogawa, Yuzu Naruse.

**Methodology:** Ayumi Goto.

**Project administration:** Ayumi Goto.

Resources: Ayumi Goto.

Software: Ayumi Goto.

Supervision: Michihiro Hashimoto, Hitoshi Yamashita.

Validation: Ayumi Goto, Hitoshi Yamashita.

Visualization: Ayumi Goto, Michihiro Hashimoto, Hitoshi Yamashita.

Writing – original draft: Ayumi Goto, Michihiro Hashimoto, Hitoshi Yamashita.

Writing – review & editing: Ayumi Goto, Michihiro Hashimoto, Hitoshi Yamashita.

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
