## [Decision Letter · Decision Letter 0]

PONE-D-25-09063Cellular repressor of E1A-stimulated genes 1 enhances skeletal muscle performance through the stimulation of muscle differentiation and fiber transition via Akt-mTOR signaling pathway activationPLOS ONE

Dear Dr. Goto,

Thank you for submitting your manuscript to PLOS ONE. After careful consideration, we feel that it has merit but does not fully meet PLOS ONE’s publication criteria as it currently stands. Therefore, we invite you to submit a revised version of the manuscript that addresses the points raised during the review process.

Your manuscript has been assessed by two experts. They have expressed interest in the manuscript but also raised several concerns. Please address all comments by point-by-point adequately.

In particular, we request a full response to the changes in grip strength.

We look forward to receiving your revised manuscript.

Kind regards,

Keisuke Hitachi

Academic Editor

PLOS ONE

“Grant-in-Aid for Scientific Research (Kakenhi) from the Japan Society for the Promotion of Science (20K06450 and 23K10881 to Hitoshi Yamashita, 21K17668 to Ayumi Goto, 18K11034 to Michihiro Hashimoto), YOKOYAMA Foundation for Clinical Pharmacology (YRY-2020 to Ayumi Goto), Naito Research Grant (Ayumi Goto), The Hibi Science Foundation (Ayumi Goto), and Chubu University Grant R (Ayumi Goto).”

“The authors would like to thank M. Matsui, Y. Yamashita, Y. Nishimoto, Y. Endo, I. Matsuda, and K. Miyake (Chubu University) for their technical support. The work was supported by Grant-in-Aid for Scientific Research (Kakenhi) from the Japan Society for the Promotion of Science (20K06450 and 23K10881 to HY, 21K17668 to AG, 18K11034 to MH), YOKOYAMA Foundation for Clinical Pharmacology (YRY-2020 to AG), Naito Research Grant (AG), The Hibi Science Foundation (AG), and Chubu University Grant R (AG).”

“Grant-in-Aid for Scientific Research (Kakenhi) from the Japan Society for the Promotion of Science (20K06450 and 23K10881 to Hitoshi Yamashita, 21K17668 to Ayumi Goto, 18K11034 to Michihiro Hashimoto), YOKOYAMA Foundation for Clinical Pharmacology (YRY-2020 to Ayumi Goto), Naito Research Grant (Ayumi Goto), The Hibi Science Foundation (Ayumi Goto), and Chubu University Grant R (Ayumi Goto).”

Reviewers' comments:

Reviewer's Responses to Questions

**Comments to the Author**

1. Is the manuscript technically sound, and do the data support the conclusions?

Reviewer #1: Yes

Reviewer #2: Partly

2. Has the statistical analysis been performed appropriately and rigorously? 

Reviewer #1: Yes

Reviewer #2: I Don't Know

3. Have the authors made all data underlying the findings in their manuscript fully available?

Reviewer #1: Yes

Reviewer #2: No

4. Is the manuscript presented in an intelligible fashion and written in standard English?

Reviewer #1: Yes

Reviewer #2: Yes

5. Review Comments to the Author

Reviewer #1: The Manuscript authored by Goto et al. aims to contribute to our understanding of CREG1's role in skeletal muscle function and differentiation, in addition to associated mechanisms. Overall, this paper is well-written and provides both in-vivo and in-vitro data to support the role of CREG1 in skeletal muscle.

As such, I will provide some minor revisions below:

1. Where appropriate, please provide more references to support claims made in the introduction and discussion. Examples include lines 51-54 when discussing sarcopenia, and lines: 374-375 when discussing the proportion of fiber types in athletes.

2. Please define all terms/names upon first use in the manuscript, e.g., line: 64, NTERA-2

3. Please add the year of publication when referencing studies by author name. E.g., "Song et al. (Year)"

4. Although soleus Myh1 expression was higher in Tg mice, it was not statistically significant as reported in the abstract. Please specify this in the abstract, results and discussion (that Myh1 trended towards higher but was not significant)

5. Along the same lines, specify that MyoD1 trended towards lower in CREG1 knockdown cells but was also not significant

6. Line 368: should this say myh7, not myh1?

7. Line 391-393: "CREG1 is thought to be mainly supplied by the liver to other organs, including skeletal muscles, exogenous CREG1 may still influence skeletal muscle function in the absence of endogenous CREG1" - modify this sentence to specify that exogenous in this context means from non-muscle tissues (e.g., liver) and endogenous means produced within skeletal muscle

8. Make a limitations section following the discussion and move the mention of the lack of muscle performance data (line 456) to this section

Reviewer #2: This manuscript by Goto et al. is aimed to determine the effect of tissue-specific CREG1 overexpression on skeletal muscle. Several points are unclear, and these should be addressed.

1) The conclusion that the increase in grip strength may be mediated by a shift in fiber type is not well supported. First, while it is true that fast-twitch fibers are often associated with greater maximal muscle strength, this is almost entirely explained by the fibers being larger. For example: PMID: 11148759, PMID: 21317219. There is no evidence provided that the muscles are larger. Indeed, in the transgenic, the soleus muscle are 15% smaller, while the EDL are 7.6% smaller (Table 1). Second, grip strength was only done with the forelimbs, which consist of almost entirely fast-twitch muscles (PMID: 22938020), like the plantaris of the hindlimb. Since the plantaris had no shift in myosin heavy chain (Fig 2), there is no rationale to presume that the myosin heavy chain composition of the forelimbs changed. Therefore, the conclusion that a shift in myosin heavy chain is mediating an increase in strength must either be supported with additional data or should be purged from this manuscript (Title, Abstract, Discussion)

2) All the individual data points must be shown for all the bar graphs. This is now common practice for rigorous presentation of data, and PLOS Data policy requires the authors to show the underlying data.

3) Figure 3. The blots and calculations for the total amount of IGF2R, Akt, and mTOR must be shown and quantified here. It is useful to know whether the increase in the phospho amount is due to 1) an increase in the fraction that is phosphorylated (i.e., strictly signaling) or 2) due to increases in the total amount of the protein with the same faction phosphorylated (i.e., not likely signaling).

4) Line 222. This is too vague to really understand which test was used when. Explicitly add which statistical analysis and comparisons were performed for each panel in the figure legends.

5) Line 37. State the test used to assess fatigue resistance. This has a major impact on interpretation of findings.

6) Line 109. Explicitly state here in the Methods the cell specificity of this transgenic. Most promoters are expressed in more than one tissue, especially at various times during development. aP2 is no exception. Add that information.

7) Line 110. Specify the breeding scheme. Were heterozygous mice breed to each other? Were the WT mice littermates of the transgenic mice, or were they from different groups?

8) Line 144. How was the myosin heavy chain staining analyzed. What software was use? Was it automated or semi-automated? Were all the fibers of every muscle analyzed, or only a sub-set?

9) Line 160. Was pen/strep used? This is a common additive to C2C12 cultures. If used, state how much? If not used, state this explicitly.

10) Line 236. Calculate and provide in the text the % difference in grip strength. Knowing the precise amount will help put this measure in the context of the % changes in myosin heavy chain.

11) Line 237-238. Was the difference in wire-hanging statistically significant? This needs to be clarified, since no indices of significance are provided in the Figure 1B.

12) Line 282. Why do the authors conclude that CREG1 is upregulated in skeletal muscle instead of adipocytes, which are in whole skeletal muscles (PMID: 32624006, PMID: 34382019)? This should be thoroughly defended or at least acknowledged in Discussion. Further, for precision, it should be stated that the upregulation is in muscle “tissue”. The current phrasing suggests it is up in muscle cells.

13) Figure 1B. Add the SEM to this panel, as it states in the figure legend.

14) Figure 2G, H. Rearrange these graphs such that the order of mRNA matches that of the proteins in the graph above each. In other words, they should be from left to right: Myh7, Myh2, Myh1, and Myh4. A reader is obviously interested in comparing the changes in protein to that of mRNA.

6. PLOS authors have the option to publish the peer review history of their article (what does this mean?). If published, this will include your full peer review and any attached files.

Reviewer #1: No

Reviewer #2: No

---

## [Author Response · Author response to Decision Letter 1]

4 Jun 2025

Reviewer #1: The Manuscript authored by Goto et al. aims to contribute to our understanding of CREG1's role in skeletal muscle function and differentiation, in addition to associated mechanisms. Overall, this paper is well-written and provides both in-vivo and in-vitro data to support the role of CREG1 in skeletal muscle.

We are thankful for the reviewers’ helpful comments that have improved the quality of our manuscript. Changes have been marked in red in the “Revised Article with Changes Highlighted” file.

As such, I will provide some minor revisions below:

1. Where appropriate, please provide more references to support claims made in the introduction and discussion. Examples include lines 51-54 when discussing sarcopenia, and lines: 374-375 when discussing the proportion of fiber types in athletes.

Thank you so much for your valuable comment. We agree with you. According to your suggestion, we have added the references (line 54 and line 395).

2. Please define all terms/names upon first use in the manuscript, e.g., line: 64, NTERA-2.

We thank the reviewer for these insightful comments. TERA2 is one of the oldest extant cell lines established from a human teratocarcinoma. NTERA-2 cells were re-established in culture from a xenograft of the TERA-2 cell line grown in a nude mouse. (PW Andrews. Human teratocarcinomas. 1988). Therefore, NTERA2 is a cell line name, not an abbreviation. We believe the current description is correct. We have checked the manuscript again to ensure that all terms/names are properly defined upon their first use.

3. Please add the year of publication when referencing studies by author name. E.g., "Song et al. (Year)"

Thank you so much for your valuable comment. We agree with you. According to your suggestion, we have added the year of publication in the manuscript (lines 79, 379, 407, 409, 440, 456 and 462).

4. Although soleus Myh1 expression was higher in Tg mice, it was not statistically significant as reported in the abstract. Please specify this in the abstract, results and discussion (that Myh1 trended towards higher but was not significant)

Thank you so much for your valuable suggestions. We agree with you. According to your suggestion, we have revised the abstract (Lines 38-39) and results (Lines 274-275). In response to the feedback from another reviewer, we have removed the discussion section.

5. Along the same lines, specify that MyoD1 trended towards lower in CREG1 knockdown cells but was also not significant

Thank you so much for your valuable comment. We agree with you. According to your suggestion, we have revised the abstract (line 42) and results (lines 338-340).

6. Line 368: should this say myh7, not myh1?

Thank you for providing these insights. We made a mistake with Myh1, and Myh7 is the correct notation. However, in accordance with the feedback from another reviewer, we have removed the relevant portion of the manuscript.

7. Line 391-393: "CREG1 is thought to be mainly supplied by the liver to other organs, including skeletal muscles, exogenous CREG1 may still influence skeletal muscle function in the absence of endogenous CREG1" - modify this sentence to specify that exogenous in this context means from non-muscle tissues (e.g., liver) and endogenous means produced within skeletal muscle.

Thank you for providing these insights. We agree with you. According to your suggestion, we have changed “CREG1 produced by non-muscle tissues may still influence skeletal muscle function in the absence of endogenous CREG1 in skeletal muscle (lines 412-414)”

8. Make a limitations section following the discussion and move the mention of the lack of muscle performance data (line 456) to this section

Thank you for providing these insights. We agree with you. According to your suggestion, we have made a limitation section (lines 481-486).

Reviewer #2: This manuscript by Goto et al. is aimed to determine the effect of tissue-specific CREG1 overexpression on skeletal muscle. Several points are unclear, and these should be addressed.

We are thankful for the reviewers’ helpful comments that have improved the quality of our manuscript. Changes have been marked in red in the “Revised Article with Changes Highlighted” file.

1) The conclusion that the increase in grip strength may be mediated by a shift in fiber type is not well supported. First, while it is true that fast-twitch fibers are often associated with greater maximal muscle strength, this is almost entirely explained by the fibers being larger. For example: PMID: 11148759, PMID: 21317219. There is no evidence provided that the muscles are larger. Indeed, in the transgenic, the soleus muscle are 15% smaller, while the EDL are 7.6% smaller (Table 1). Second, grip strength was only done with the forelimbs, which consist of almost entirely fast-twitch muscles (PMID: 22938020), like the plantaris of the hindlimb. Since the plantaris had no shift in myosin heavy chain (Fig 2), there is no rationale to presume that the myosin heavy chain composition of the forelimbs changed. Therefore, the conclusion that a shift in myosin heavy chain is mediating an increase in strength must either be supported with additional data or should be purged from this manuscript (Title, Abstract, Discussion)

We appreciate the reviewer’s thoughtful comments and have carefully considered the points raised. In accordance with your suggestion, we have removed the conclusion that a shift in myosin heavy chain mediates the increase in strength from the manuscript. According to your suggestion, we have revised title (lines 3-4), abstract (lines 45-46) and discussion section (lines 386-391).

2) All the individual data points must be shown for all the bar graphs. This is now common practice for rigorous presentation of data, and PLOS Data policy requires the authors to show the underlying data.

Thank you so much for your valuable comment on our study. According to your suggestion, we have revised the graph to all the individual data points.

3) Figure 3. The blots and calculations for the total amount of IGF2R, Akt, and mTOR must be shown and quantified here. It is useful to know whether the increase in the phospho amount is due to 1) an increase in the fraction that is phosphorylated (i.e., strictly signaling) or 2) due to increases in the total amount of the protein with the same faction phosphorylated (i.e., not likely signaling).

Thank you so much for your valuable comment on our study. According to your suggestion, we have added the results (lines 307-308), Fig 3D and 3E.

4) Line 222. This is too vague to really understand which test was used when. Explicitly add which statistical analysis and comparisons were performed for each panel in the figure legends.

Thank you so much for your valuable comment on our study. According to your suggestion, we have revised the Statistical analyses section and added the statistical analysis in the figure legends.

5) Line 37. State the test used to assess fatigue resistance. This has a major impact on interpretation of findings.

Thank you so much for your valuable comment on our study. According to your suggestion, we have revised the abstract (line 37).

6) Line 109. Explicitly state here in the Methods the cell specificity of this transgenic. Most promoters are expressed in more than one tissue, especially at various times during development. aP2 is no exception. Add that information.

Thank you for pointing this out. Unfortunately, we have not been able to investigate cell specificity in our transgenic mouse model. It has been previously demonstrated that aP2/FABP4 is not only expressed in adipocytes but also macrophages (Makowski L, et al. 2005, Yang J, et al. 2023). According to your suggestion, we have added the methods (Lines 109-110).

7) Line 110. Specify the breeding scheme. Were heterozygous mice breed to each other? Were the WT mice littermates of the mice, or were they from different groups?

Thank you so much for your valuable comment on our study. We crossed the heterozygous transgenic mice (Tg; 14 th generation) with their wild-type (WT) littermates, and used the resulting Tg mice and their WT littermates for the study. According to your suggestion, we have added the methods (lines 110-113).

8) Line 144. How was the myosin heavy chain staining analyzed. What software was use? Was it automated or semi-automated? Were all the fibers of every muscle analyzed, or only a sub-set?

We thank the reviewer for these insightful comments. Samples were visualized on a microscope and analyzed using ImageJ software in the IHC staining. All stained fibers in the muscle cross-section were counted manually. Previous studies have demonstrated that the number of fibers was counted and expressed as the proportion of the total number of fibers stained (Shibaguchi T, et al. 2019). Therefore, it can be considered that there is no issue with this method. According to your suggestion, we have added the methods (lines 162-164).

9) Line 160. Was pen/strep used? This is a common additive to C2C12 cultures. If used, state how much? If not used, state this explicitly.

Thank you so much for your valuable comment on our study. According to your suggestion, we have revised the methods for C2C12 cell culture (lines 172-173).

10) Line 236. Calculate and provide in the text the % difference in grip strength. Knowing the precise amount will help put this measure in the context of the % changes in myosin heavy chain.

We thank the reviewer for these insightful comments. In response to your suggestion, we have added the text on the percentage difference in grip strength (line 242).

11) Line 237-238. Was the difference in wire-hanging statistically significant? This needs to be clarified, since no indices of significance are provided in the Figure 1B.

We thank the reviewer for these insightful comments. As you pointed out, no statistical analysis was performed on the graph in Fig. 1B. In the wire-hanging test, the number of falls is used as the score without statistical processing. In fact, no statistical processing was performed in the wire-hanging test in the paper by Egawa T, et al. (2017) and Raymackers JM, et al. (2003). Therefore, we believe that the absence of statistical analysis in this graph is not a problem.

12) Line 282. Why do the authors conclude that CREG1 is upregulated in skeletal muscle instead of adipocytes, which are in whole skeletal muscles (PMID: 32624006, PMID: 34382019)? This should be thoroughly defended or at least acknowledged in Discussion. Further, for precision, it should be stated that the upregulation is in muscle “tissue”. The current phrasing suggests it is up in muscle cells.

Thank you so much for your valuable comment on our study. We were unable to elucidate the localization of skeletal muscle CREG1 by immunofluorescence staining in this study. However, we did not observe any intramuscular lipid droplets in the histological analysis. Therefore, we believe that the influence of CREG1-overexpressing adipocyte on total CREG1 expression was minimal, if any, in Tg skeletal muscle. So, we think that most of CREG1 protein detected in muscle lysates was that intracellularly produced in the muscle cells. We discussed the line　411-434. Additionally, exogenous CREG1 derived from other tissues via blood stream can be taken into muscle cells (Kusudo 2022), as we detected its processed form (~20 kD) (Fig. 3C). According to your suggestion, we have changed the “CREG1 upregulation in skeletal muscle enhances Akt-mTOR signaling” to “CREG1 upregulation in skeletal muscle tissue enhances Akt-mTOR signaling (line 292)”.

13) Figure 1B. Add the SEM to this panel, as it states in the figure legend.

We thank the reviewer for these insightful comments. As described in the methods, “Data are expressed as the average fall score, with each mouse starting with a score of 10, which was reduced by 1 after each fall. The average fall score at a given time during the test was calculated using the following equation: (10n−x)/n, where n is the number of animals and x is the cumulative number of falls.” Therefore, the average fall score is calculated by scoring the number of falls for each mouse, meaning there is no standard error. The papers by Egawa T, et al. (2017) and Raymackers JM et al. (2003), both of whom conducted the wire-hanging test, also do not report SEM. Accordingly, I am confident that the current graph is acceptable.

14) Figure 2G, H. Rearrange these graphs such that the order of mRNA matches that of the proteins in the graph above each. In other words, they should be from left to right: Myh7, Myh2, Myh1, and Myh4. A reader is obviously interested in comparing the changes in protein to that of mRNA.

Thank you so much for your valuable comment on our study. We agree with you. According to your suggestion, we have revised Figure 2G, 2H, results (line 272-273) and figure legends (line 287).

---

## [Decision Letter · Decision Letter 1]

PONE-D-25-09063R1Cellular repressor of E1A-stimulated genes 1 enhances skeletal muscle performance through the stimulation of muscle differentiation and Akt-mTOR signaling pathway activationPLOS ONE

Dear Dr. Goto,

Thank you for submitting your manuscript to PLOS ONE. After careful consideration, we feel that it has merit but does not fully meet PLOS ONE’s publication criteria as it currently stands. Therefore, we invite you to submit a revised version of the manuscript that addresses the points raised during the review process.

Thank you for revising your paper. However, there still seem to be some minor concerns. Please revise your manuscript according to the reviewers' comments before publication.==============================

We look forward to receiving your revised manuscript.

Kind regards,

Keisuke Hitachi

Academic Editor

PLOS ONE

Journal Requirements:

Reviewers' comments:

Reviewer's Responses to Questions

**Comments to the Author**

1. If the authors have adequately addressed your comments raised in a previous round of review and you feel that this manuscript is now acceptable for publication, you may indicate that here to bypass the “Comments to the Author” section, enter your conflict of interest statement in the “Confidential to Editor” section, and submit your "Accept" recommendation.

Reviewer #2: (No Response)

2. Is the manuscript technically sound, and do the data support the conclusions?

Reviewer #2: Yes

3. Has the statistical analysis been performed appropriately and rigorously? 

Reviewer #2: No

4. Have the authors made all data underlying the findings in their manuscript fully available?

Reviewer #2: Yes

5. Is the manuscript presented in an intelligible fashion and written in standard English?

Reviewer #2: Yes

6. Review Comments to the Author

Reviewer #2: This reviewer appreciates the revisions to the manuscript. However, one issue remains -- analysis of the wire hanging test. If statistics cannot be performed on the data in Fig 1B, then it is not appropriate to conclude that the WT is different than Tg group. By convention in rigorous scientific literature, quantitative data is not considered different unless it is statistically different. Therefore, this data should additionally be presented in a manner where proper statistics can be calculated, for example as was done in Ref #23, Egawa et al. Then, conclusions throughout the manuscript regarding the wire hang test must be based only on data where statistics are performed.

7. PLOS authors have the option to publish the peer review history of their article (what does this mean?). If published, this will include your full peer review and any attached files.

Reviewer #2: No

---

## [Author Response · Author response to Decision Letter 2]

30 Jun 2025

We thank the reviewer for these insightful comments. We used the analysis methods from previous study (Raymackers JM, et al. (2003)) to present Figure 1b. Following your suggestion, we reanalyzed the data using Egawa et al.’s method and found no significant differences in the longest holding time or holding impulse. Therefore, we have removed Figure 1b and the description of wire hanging in the text. Changes have been marked in red in the “Revised Article with Changes Highlighted” file.

---

## [Editor Report · Decision Letter 2]

Cellular repressor of E1A-stimulated genes 1 enhances skeletal muscle performance through the stimulation of muscle differentiation and Akt-mTOR signaling pathway activation

PONE-D-25-09063R2

Dear Dr. Goto,

We’re pleased to inform you that your manuscript has been judged scientifically suitable for publication and will be formally accepted for publication once it meets all outstanding technical requirements.

Kind regards,

Keisuke Hitachi

Academic Editor

PLOS ONE

Additional Editor Comments (optional):

In the PDF, Figure 1 was located at the end of the figures, so please pay attention to the order of the figures in subsequent proofs, etc.
---

## [Editor Report · Acceptance letter]

PONE-D-25-09063R2

PLOS ONE

Dear Dr. Goto,

I'm pleased to inform you that your manuscript has been deemed suitable for publication in PLOS ONE. Congratulations! Your manuscript is now being handed over to our production team.

Kind regards,

on behalf of

Dr. Keisuke Hitachi

Academic Editor

PLOS ONE